# Quantifying Yeast Microtubules and Spindles Using the Toolkit for Automated Microtubule Tracking (TAMiT)

**DOI:** 10.3390/biom13060939

**Published:** 2023-06-04

**Authors:** Saad Ansari, Zachary R. Gergely, Patrick Flynn, Gabriella Li, Jeffrey K. Moore, Meredith D. Betterton

**Affiliations:** 1Department of Physics, University of Colorado Boulder, Boulder, CO 80309, USA; saad.ansari@colorado.edu (S.A.); zachary.gergely@colorado.edu (Z.R.G.); patrick_flynn@g.harvard.edu (P.F.); 2Department of Molecular, Cellular and Developmental Biology, University of Colorado Boulder, Boulder, CO 80309, USA; 3Department of Cell and Developmental Biology, University of Colorado School of Medicine, Aurora, CO 80045, USA; gabriella.li@cuanschutz.edu (G.L.); jeffrey.moore@cuanschutz.edu (J.K.M.)

**Keywords:** microtubule tracking, fluorescent microscopy, curve optimization, image analysis

## Abstract

Fluorescently labeled proteins absorb and emit light, appearing as Gaussian spots in fluorescence imaging. When fluorescent tags are added to cytoskeletal polymers such as microtubules, a line of fluorescence and even non-linear structures results. While much progress has been made in techniques for imaging and microscopy, image analysis is less well-developed. Current analysis of fluorescent microtubules uses either manual tools, such as kymographs, or automated software. As a result, our ability to quantify microtubule dynamics and organization from light microscopy remains limited. Despite the development of automated microtubule analysis tools for in vitro studies, analysis of images from cells often depends heavily on manual analysis. One of the main reasons for this disparity is the low signal-to-noise ratio in cells, where background fluorescence is typically higher than in reconstituted systems. Here, we present the Toolkit for Automated Microtubule Tracking (TAMiT), which automatically detects, optimizes, and tracks fluorescent microtubules in living yeast cells with sub-pixel accuracy. Using basic information about microtubule organization, TAMiT detects linear and curved polymers using a geometrical scanning technique. Images are fit via an optimization problem for the microtubule image parameters that are solved using non-linear least squares in Matlab. We benchmark our software using simulated images and show that it reliably detects microtubules, even at low signal-to-noise ratios. Then, we use TAMiT to measure monopolar spindle microtubule bundle number, length, and lifetime in a large dataset that includes several *S. pombe* mutants that affect microtubule dynamics and bundling. The results from the automated analysis are consistent with previous work and suggest a direct role for CLASP/Cls1 in bundling spindle microtubules. We also illustrate automated tracking of single curved astral microtubules in *S. cerevisiae*, with measurement of dynamic instability parameters. The results obtained with our fully-automated software are similar to results using hand-tracked measurements. Therefore, TAMiT can facilitate automated analysis of spindle and microtubule dynamics in yeast cells.

## 1. Introduction

Automated image analysis for fluorescently labeled proteins that appear as Gaussian spots have been developed and widely used for detection and tracking [1,2,3,4]. Extended protein assemblies, however, are more challenging to analyze. Given the importance of higher-order protein assemblies, improved tools would lead to a better quantification of their structure and properties. For example, microtubules, which are linear polymers made of tubulin dimer subunits play essential roles in eukaryotic cells during mitosis [5,6], intracellular transport [7], cell motility [8,9], morphogenesis [10], and axonal transport [11]. They are also a key drug target for cancer and malaria treatment [12,13]. Microtubules with fluorescent labels on tubulin dimers appear in images as Gaussian lines rather than spots, meaning that tools for their automated analysis must work differently. There has been considerable development in tools for microtubule tracking both for systems in vitro [14,15,16,17], and in cells [18,19,20,21]. However, fully automated tools for microtubule tracking in 3-dimensional (3D) live cell data are lacking. In fact, many cell biologists continue to track microtubules and proteins that bind to them manually and with kymographs. Automating microtubule tracking in living cells has proven challenging because the image signal-to-noise ratio (SNR) of these polymers can be low. Despite this technical difficulty, tracking microtubules in cells is important to understand their biological function.

One class of methods to quantify cellular microtubule dynamics is based on fluorescently tagged tip-tracking proteins. Tip tracking MAPs associate with microtubule plus-ends [22,23,24] and appear as Gaussian spots. Therefore, they can be quantified with automated particle tracking approaches [2,25], allowing quantification of microtubule dynamics [26,27,28]. However, tip-tracking proteins analyzed to date bind only to growing plus-ends, not to paused or shrinking plus-ends. Therefore, this approach suffers the disadvantage that fluorescent tracks show only microtubule growth events; catastrophe, rescue, and depolymerization dynamics must be inferred. Alternatively, when tubulin is fluorescently labeled, the fluorescence is associated directly with the microtubule lattice and does not depend on the binding of MAPs. In this case, entire microtubules can be visualized throughout their polymerization cycle, making it easy to extract structural properties, like curvature, and the kinetic parameters that describe microtubule dynamics.

Here we present a Toolkit for Automated Microtubule Tracking (TAMiT). This package automatically detects and tracks entire microtubules or microtubule bundles in yeast cells from 3D confocal fluorescence microscopy. The strength of our software lies in its detection routine coupled with a robust optimization process, leading to sub-pixel accuracy in the measurement of microtubule length. TAMiT’s object-oriented framework, focusing on inheritance-based specialization, allows users to adapt the software for their individual needs easily. As a result, TAMiT has machine-learning capabilities since it can be used to analyze other composite microtubule structures beyond those presented here. Here, we present illustrative models that we created for use with a variety of cellular phenotypes in yeasts, and we show how TAMiT uses them to detect microtubules. We benchmark our software using simulated data and show that the software performs robustly even at low SNR. Specifically, TAMiT is used to detect monopolar, bipolar, and anaphase spindles in *S. pombe*, along with the mitotic spindle in *S. cerevisiae*. In addition, TAMiT can track single astral microtubules in cells, which has proven difficult in the past. TAMiT can accurately detect both straight and curved microtubules, as shown here. Finally, we extract microtubule dynamic instability measurements in *S. cerevisiae* and compare them with hand-tracked data.

## 2. Materials and Methods

The design of TAMiT is based on key features of microtubule assemblies in yeast cells. First, microtubules in cells are often present in dynamic higher-order structures [29]. For example, in *S. pombe* and *S. cerevisiae* mitotic spindles, microtubule minus ends are anchored near spindle-pole bodies (SPBs) via γ-tubulin complexes [30]. In a bipolar spindle, two SPBs are linked by a bundle of microtubules that appears as a diffraction-limited line, while a monopolar spindle has unseparated SPBs with lines emanating outward (Figure 1). A single SPB in yeasts can anchor dozens of microtubules. Because many of these microtubules are short, they appear as point sources of tubulin fluorescence (spots). Our software learns an underlying mathematical model for fluorescence distribution. This model is optimized by non-linear fitting such that the fitted intensity matches the original image. Below, we describe the mathematical models for all the features, followed by a discussion of detection, optimization, and tracking. Finally, we present the results of the validation.

### 2.1. Mathematical Model

To achieve subpixel accuracy in microtubule detection, each feature that comprises an image must be represented by an appropriate mathematical model. The image can then be represented by a combination of these features. TAMiT currently supports three basic features (Figure 2A and Appendix A). A *spot* of chosen intensity and width represents a Gaussian distribution (a point source object), which may be smaller than the resolving power of the microscope, such as an assembly of short microtubules emanating from an SPB. We use a spot to represent the SPB (the position which coincides with the position of the spindle microtubule minus-ends). A *line* represents a single microtubule or a bundle that is straight and longer than a few image pixels (typically, a microtubule must be at least a few hundred nanometers long to be visible as a line). These appear as a line with intensity represented by an integrated Gaussian distribution. We use the line to model microtubules in cells where the polymers are not bent. A *curve* represents a single or a bundle of microtubules that is curved. The fluorescence distribution of a curved microtubule has a curved centerline with diffraction-limited fluorescence along that line. These basic features can be combined to construct structures such as either a monopolar or a bipolar spindle (Figure 2B).

#### 2.1.1. Spot

For a collection of short microtubules at some position ${\vec{x}}_{0}=\left(x_{0},y_{0},z_{0}\right)$, the intensity appears as a point Gaussian with variable width. We model this as a spot with amplitude *A*, centered at position ${\vec{x}}_{0}=\left(x_{0},y_{0},z_{0}\right)$, and having Gaussian width $\vec{\sigma }=\left(\sigma_{x},\sigma_{y},\sigma_{z}\right)$. Therefore, seven parameters must be fit for every spot. These are $\theta =\{A,{\vec{x}}_{0},\vec{\sigma }\}$. The Gaussian intensity of a spot at probe position $\vec{x}=\left(x,y,z\right)$ is calculated as
(1)$$f_{\theta }\left(x,y,z\right)=Ae^{-{\left(\frac{x-x_{0}}{\sigma_{x}}\right)}^{2}}e^{-{\left(\frac{y-y_{0}}{\sigma_{y}}\right)}^{2}}e^{-{\left(\frac{z-z_{0}}{\sigma_{z}}\right)}^{2}}$$

#### 2.1.2. Line

We model a straight microtubule or bundle as a Gaussian line of amplitude *A*, start position ${\vec{x}}_{0}=\left(x_{0},y_{0},z_{0}\right)$, end position ${\vec{x}}_{1}=\left(x_{1},y_{1},z_{1}\right)$ and Gaussian width $\vec{\sigma }=\left(\sigma_{x},\sigma_{y},\sigma_{z}\right)$. We adopt a parametric representation for coordinates along the line, ${\vec{x}}_{i}\left(t\right)=\left(x_{i}\left(t\right),y_{i}\left(t\right),z_{i}\left(t\right)\right)$ where
$$\begin{array}{cc} x_{i}\left(t\right) & =x_{0}+\left(x_{1}-x_{0}\right)t\\ y_{i}\left(t\right) & =y_{0}+\left(y_{1}-y_{0}\right)t\\ z_{i}\left(t\right) & =z_{0}+\left(z_{1}-z_{0}\right)t \end{array}$$
for t∈[0,1] such that ${\vec{x}}_{i}\left(t=0\right)=\left(x_{0},y_{0},z_{0}\right)$ and ${\vec{x}}_{i}\left(t=1\right)=\left(x_{1},y_{1},z_{1}\right)$. We fit ten parameters for every line: $\theta =\{A,{\vec{x}}_{0},{\vec{x}}_{1},\vec{\sigma }\}$. The contribution of the Gaussian intensity $f_{\theta }\left(x,y,z\right)$ at position $\vec{x}=\left(x,y,z\right)$ is then given by an integral of point Gaussians along the normalized arc length *t*: (2)$$f_{\theta }\left(x,y,z\right)=\int_{t=0}^{t=1}dtAe^{-{\left(\frac{x-x_{i}\left(t\right)}{\sigma_{x}}\right)}^{2}}e^{-{\left(\frac{y-y_{i}\left(t\right)}{\sigma_{y}}\right)}^{2}}e^{-{\left(\frac{z-z_{i}\left(t\right)}{\sigma_{z}}\right)}^{2}}$$
We can expand the exponentials in the integrand to obtain
$$\begin{array}{cc} f_{\theta }\left(x,y,z\right) & =\int_{t=0}^{t=1}dtA\exp\left(-B-2tC\left(t\right)-t^{2}D\left(t\right)\right)\\ B & =\frac{x^{2}}{\sigma_{x}^{2}}+\frac{y^{2}}{\sigma_{y}^{2}}+\frac{z^{2}}{\sigma_{z}^{2}}\\ C(t) & =\frac{xx_{i}\left(t\right)}{\sigma_{x}^{2}}+\frac{yy_{i}\left(t\right)}{\sigma_{y}^{2}}+\frac{zz_{i}\left(t\right)}{\sigma_{z}^{2}}\\ D(t) & =\frac{x_{i}^{2}\left(t\right)}{\sigma_{x}^{2}}+\frac{y_{i}^{2}\left(t\right)}{\sigma_{y}^{2}}+\frac{z_{i}^{2}\left(t\right)}{\sigma_{z}^{2}} \end{array}$$
While an integral in this form can be calculated numerically, we can also evaluate it using error functions.

#### 2.1.3. Curve

In certain situations, curved or bending microtubules may be present. To develop a curve model, we assume that the microtubule/bundle centerline follows a parametric polynomial in all three dimensions.
$$\begin{array}{cc} x(t) & =a_{0}+a_{1}t+...+a_{n}t^{n}\\ y(t) & =b_{0}+b_{1}t+...+b_{n}t^{n}\\ z(t) & =c_{0}+c_{1}t+...+c_{n}t^{n} \end{array}$$
While the choice of polynomial order is arbitrary for this model, here, we assume that n=4 in the *X* and *Y* dimensions (where the pixel size is smallest) and n=1 in the *Z* dimension (where the number of pixels is limited, so selecting a higher order can be impossible). With this assumption, curves seen in the XY plane are linear in the *Z* dimension. We calculate a local curvature for the microtubule using:$$\begin{array}{cc} \vec{K}\left(t\right) & =\frac{x^{\prime }\left(t\right)y^{\prime \prime }\left(t\right)-y^{\prime }\left(t\right)x^{\prime \prime }\left(t\right)}{\sqrt{{\left(x^{\prime }{\left(t\right)}^{2}+y^{\prime }{\left(t\right)}^{2}\right)}^{3}}} \end{array}$$
where $x^{\prime }\left(t\right)$ and $y^{\prime }\left(t\right)$ are first-order time derivatives and $x^{\prime \prime }\left(t\right)$ and $y^{\prime \prime }\left(t\right)$ second-order. We generate a model for this curvature using a Fourier function
(3)$$\begin{array}{c} \tilde{K}\left(t\right)=k_{0}+\sum_{i>0}^{N}p_{i}\cos\left(iwt\right)+q_{i}\sin\left(iwt\right) \end{array}$$
where we fit $\theta_{k}=\left\{k_{0},w,p_{1},...p_{n},q_{1},...,q_{N}\right\}$. This forms the initial estimate for the microtubule curvature. Given a start point ${\vec{x}}_{0}=\left(x_{0},y_{0},z_{0}\right)$, an initial tangent vector $\vec{T}\left(t=0\right)$, the curvature coefficients $\theta_{k}$, and a microtubule length, we can then find all points along the microtubule. We do this by iteratively computing the tangent (Equation (Equation 4)) and the coordinates (Equation (Equation 5)) along the curve. Here $\bar{R}$ is a 3 × 3 matrix that produces a π/2 rotation in the XY direction, such that $\bar{R}\hat{T}\left(t\right)$ gives the direction of the normal vector for any t.
(4)$$\begin{array}{cc} \vec{T}\left(t+dt\right) & =\vec{T}\left(t\right)+\left(\bar{R}\hat{T}\left(t\right)\right)\tilde{K}\left(t\right)dt \end{array}$$
(5)$$\begin{array}{cc} \vec{x}\left(t+dt\right) & =\vec{x}\left(t\right)+dt\vec{T}\left(t\right) \end{array}$$
For the data in this paper, we found that using a Fourier series approximation with N=2 was adequate. Using N=1 gave a visibly poor fit, while for N>1, the fit residual had a similar value, and the fit visibly looked similar. For application to other curved microtubules, checking the fit residual as *N* is varied can be used to select the best value of *N*. With that selection, there are 15 parameters for every curve, namely $\theta =\{A,{\vec{x}}_{0},\vec{T}\left(t=0\right),\theta_{k},\vec{\sigma }\}$. The Gaussian intensity $f_{\theta }\left(x,y,z\right)$ at $\vec{x}=\left(x,y,z\right)$ is calculated by numerical integration via Gauss quadrature [31].

### 2.2. Detection

The first step in analyzing an image is the detection process, which generates an initial guess of features and their location to be used in fitting the full model. TAMiT starts by applying a 3D Gaussian filter to enhance the tubulin intensity and smooth out variation from noise. The next steps in detection are then specific to the particular tubulin structure being modeled. For example, a *S. pombe* monopolar spindle (center row Figure 1A,B) contains a bright central point at the location of the single SPB or two adjacent SPBs. TAMiT models this as a single 3D Gaussian spot. To find this spot, we use Otsu’s method for thresholding [32] followed by an Extended-maxima H-tranform [33] on the MIP along the *z*-axis of the image. The brightest pixel then corresponds to the position of the SPB/SPBs (Appendix A). There are also microtubules emanating from the SPB. To find these, TAMiT transforms the MIP image to a polar representation I(r,ϕ) with the SPB at the origin and radially integrates the intensity to get an angular intensity function I(ϕ) (Appendix A) The locations of peaks $\phi_{i}$ in I(ϕ) correspond to the angular coordinates of possible lines. The length coordinate $L_{i}$ is determined by fixing the angle and increasing the radius until the polar intensity function $I(r,\phi =\phi_{i})$ becomes comparable to the background intensity. TAMiT also implements a minimum and maximum length cutoff for lines.

A bipolar spindle (top row Figure 1A,B) contains a collection of microtubules between two spindle pole bodies (SPBs). This collection of microtubules appears as a single bright Gaussian line, and we model it as a single line. This line connects two SPBs, which are modeled as spots. To find the spindle, TAMiT uses Otsu’s thresholding followed by an Extended-maxima H-tranform on the MIP image to find a bright spindle region (Appendix A). Connected component analysis can then be used to find an orientation vector for the bright region. We find two maximally distant end-points with high intensity along the orientation vector (Appendix A). These two endpoints correspond to the two SPBs. To find lines emanating from each SPB, we use the same technique as that described for monopolar spindles. To find curves emanating from an SPB, we start by finding the initial direction of the curve. Next, steerable filtering is used to enhance pixels through which the curve passes [34]. TAMiT iteratively steps along the brightest pixels, finding a new local orientation at each step. The iteration stops once the intensity drops below a threshold set by the background (Appendix A).

Once all features and their models have been found, the full model is given by the sum of the background fluorescence intensity $B_{0}$ and all the individual feature models:(6)$$\begin{array}{c} F_{\theta }\left(x,y,z\right)=B_{0}+\sum f_{\theta }\left(x,y,z\right) \end{array}$$

### 2.3. Optimization

To better represent the image and measure microtubule length with sub-pixel accuracy, we fit the model parameters. The optimization in TAMiT is performed using non-linear least squares fitting, in which we minimize the residual, the sum of squared differences between the model and experimental images. For voxel coordinates x,y,z, and image intensity $I_{0}\left(x,y,z\right)$, the optimized parameters are
$$\hat{\theta }=\underset{\theta }{\mathrm{argmin}}\sum_{x,y,z}{\left(F_{\theta }\left(x,y,z\right)-I_{0}\left(x,y,z\right)\right)}^{2}$$
where $F_{\theta }$ is the model function that simulates Gaussian intensity given feature parameters θ. TAMiT first optimizes the features locally, i.e., only varying parameters specific to a single feature while keeping the parameters of other features fixed. This is followed by global optimization, where all parameters (including a background intensity level) are varied within reasonable bounds in search of a global minimum.

Lastly, TAMiT optimizes feature number, a hyperparameter. This is done by first increasing the number of lines/curves in the model until the decrease in residual is insignificant compared to the increase in the number of parameters. Next, the number of lines/curves is decreased until the increase in residual becomes significant compared to the decrease in the number of parameters. An f-test is employed for the significance criterion.

### 2.4. Tracking

After detection is performed for all time-frames in a movie, TAMiT tracks the features using a modified version of U-track [2]. This ensures that spuriously detected microtubules are ignored, and only features that existed for some minimum number of frames are kept.

### 2.5. Validation

To test the accuracy of TAMiT, we simulated and fitted monopolar spindles similar to those found experimentally in *S. pombe*. We generated 1000 3D images, based on random points in the parameter space, at varying SNR (Figure 3A,B). The SNR was defined as the mean intensity of a single microtubule divided by the background intensity (defined as the median intensity of the image). For this test, the amplitude of the SPB and each microtubule was held fixed while noise was varied. Next, we ran TAMiT on each simulated image to extract the optimized parameters and compared them to the parameters originally used for image creation. For each detected microtubule, we used a cost criterion to determine if the microtubule found was present in the simulated image. Microtubules satisfying the criterion were deemed correctly detected, while those failing were deemed spuriously detected. For example, if the end position of a detected microtubule was more than 0.5 µm away from the end position of all simulated microtubules, the detected microtubule was deemed spurious. Microtubules in the simulated data that were not found by TAMiT were labeled as missed detections. For correctly detected microtubules, we computed the error in the *X* and *Y* dimension (pixel size 0.1 µm) and in the *Z* dimension (pixel size 0.5 µm.) (Figure 3C,D). Our multiple trials show that the mean 3D error (Figure 3E) became sub-pixel for all cases where SNR >1.5. The percentage of correct detections was also above 90% for SNR >1.25 (Figure 3F). For very low SNR, the percentage of spuriously detected microtubules was high, but it became negligible for SNR >1.25 (Figure 3G). Finally, we quantified the effect of microtubule length on detection status (Figure 3H). We found that with increasing SNR, missed microtubules were more likely to be short. There was a transition at SNR =1.25. Based on these data, we conclude that TAMiT is accurate at detecting straight microtubules at SNR >1.25 and achieves sub-pixel accuracy at SNR >1.5.

### 2.6. Experimental Methods

#### 2.6.1. *S. pombe*

*S. pombe* strains (Appendix A) were cultured using standard techniques [35]. Strains were constructed using genetic crosses and random spore analysis to isolate genotypes of interest. All microscopy images and datasets were obtained using live cell preparation. Cells were grown on YES plates and imaged in EMM liquid media to reduce background fluorescence. Bipolar spindles were imaged at 25 ${}^{{}^{\circ}}$C, and monopoles were imaged at 37 ${}^{{}^{\circ}}$C. To obtain sufficient monopolar spindles, *cut7-24* cells were placed at 37 ${}^{{}^{\circ}}$C for 2–4 h and then imaged at this restrictive temperature. Cells were transferred from a 37 ${}^{{}^{\circ}}$C incubator to the microscope in less than 30 seconds to prevent the temperature-driven transition from monopolar to bipolar spindles. The temperature was maintained with $\pm 0.1{\;}^{{}^{\circ}}$C using a CherryTemp temperature controller (Cherry Biotech, Rennes, France). Spinning-disk confocal microscopy was performed on a Nikon Eclipse Ti microscope described previously [36,37,38]. The fluorescent label for microtubules was obtained by expressing a mCherry-α-tubulin-chimera at a low level (∼10% wild type α-tubulin), as described previously [37,38,39]. The low-level tubulin labeling helps reduce tag-related perturbations to microtubule dynamics. 3D time-lapse images were obtained using the EM Gain laser settings on the Nikon illumination system and the number of Z-planes indicated previously [38]. Detailed strain information is provided in the Appendix A.

#### 2.6.2. *S. cerevisiae*

Budding yeast was grown in standard media and then manipulated and transformed by standard methods [40]. GFP-Tub1 fusions were integrated into the genome and expressed ectopically, in addition to the native α-tubulin genes TUB1 and TUB3 [41]. We estimate that GFP-Tub1 comprises approximately 25% of the total α-tubulin expressed in these cells [42]. Cells were grown asynchronously to the early log phase in a nonfluorescent medium and adhered to slide chambers coated with concanavalin A [43]. Images were collected on a Nikon Ti-E microscope equipped with a 1.45 NA 100× CFI Plan Apo objective, piezoelectric stage (Physik Instrumente; Auburn, MA, USA), spinning disk confocal scanner unit (CSU10; Yokogawa, Musashino, Tokyo), 488 nm laser (Agilent Technologies; Santa Clara, CA, USA), and an EMCCD camera (iXon Ultra 897; Andor Technology; Belfast, UK) using NIS Elements software (Nikon, Minato City, Tokyo). During imaging, sample temperature was maintained at $37{\;}^{{}^{\circ}}$C as indicated using the CherryTemp system (CherryBiotech; Rennes, France). Z-stacks consisting of 12 images separated by 0.45 µm were collected at 5 second intervals for 10 minutes. All analyses were conducted in pre-anaphase cells, which typically exhibit one or two individual astral microtubules extending from each SPB [44].

### 2.7. Manual Analysis of Microtubule Dynamics in S. cerevisiae

Astral microtubule lengths were measured in each maximum intensity projection (2D data), beginning at the cytoplasmic edge of the SPB to the tip of the astral microtubule; therefore, any displacement of the SPB does not impact microtubule length measurement. Assembly and disassembly events were defined as at least three contiguous data points that produced a length change ≥0.5 µm with a coefficient of determination ≥0.8. The length of polymerization before a catastrophe event was calculated by determining the total length of polymerization before a switch to depolymerization.

## 3. Results

Microtubule structures from different stages of the cell cycle pose different challenges for the task of feature detection. TAMiT uses specialized models for the individual cases shown in Figure 4. The bipolar mitotic spindle in *S. pombe* is visible as a bright line of microtubule fluorescence that may be accompanied by some much fainter polar microtubules that project from the end(s) of the spindle. Figure 4A shows the tracking of a bipolar spindle over time. At the third frame shown, a single polar microtubule first appears and is detected and tracked by TAMiT as it changes length. Mutations of kinesin-5/Cut7 in *S. pombe* can lead to monopolar spindles when the SPBs do not separate (Figure 4B). Here in the temperature-sensitive *cut7-24*, dynamic microtubule bundles can rotate about their attachment points at the SPB. Anaphase spindle elongation in *S. pombe* leads to changes in spindle length over time (Figure 4C), which the software accurately captures. TAMiT can also find and track microtubules in *S. cerevisiae*. Here, long curved astral microtubules can grow and curve (Figure 4D). These are primarily single microtubules but can sometimes be a bundle of two microtubules [43].

### 3.1. Quantification of Monopolar Spindle Microtubule Number, Length, and Lifetime

Mutations in the *S. pombe* genome can perturb microtubule behavior and lead to abnormal spindle structures that provide insights into the mechanisms of mitosis. For example, mutations in the kinesin-5/Cut7 motor can prevent assembly of the bipolar spindle [45]. In these cells, SPBs do not separate, and a monopolar spindle forms with microtubules whose plus-ends point radially away from the central spindle pole. Because monopolar spindles arrest cells in mitosis, they can be used to assess how other perturbations alter mitotic microtubule number, length, and bundling [36]. Therefore, we used TAMiT to study monopolar spindle microtubules in *S. pombe* cells carrying the temperature-sensitive mutant *cut7-24* [46].

In addition to the *cut7-24* reference, we additionally considered 3 perturbations that affect microtubule dynamics and bundling in mitosis. First, we added the deletion of *klp6*, a kinesin-8 motor. Because Klp6p destabilizes microtubules, in its absence, microtubules become more stable and longer [37,47,48,49,50]. We used this strain to assess whether TAMiT could identify monopolar spindle microtubule bundles in *klp*6Δ that are longer and more stable. Next, we additionally deleted *alp14*. Alp14p is a TOG/XMAP215 homolog, a microtubule plus-end tip tracking protein that promotes microtubule polymerization [51,52]. Therefore, *alp*14Δ cells contain shorter microtubules and shorter bundle lengths. Third, we additionally added the temperature-sensitive mutant *cls*1-36. Cls1p is a CLASP homolog that helps stabilize microtubules by promoting rescue [53,54,55]. The precise mechanism and interactions of Cls1p in fission yeast have remained unclear, with some work suggesting that it is recruited by the crosslinking protein Ase1p [53], while other work has found that Cls1p promotes microtubule bundling even in the absence of *ase1* [55].

We have used TAMiT to fit 3D images and detect microtubules from these strains (Figure 5A,B). Because of the automation enabled by TAMiT, we analyzed a large number of images. For example, we use 140 time points from approximately 100 cells for the *cut7-24* reference cells, a total of $10^{4}$ images. To characterize differences between the strains, we measured monopolar spindle microtubule/bundle number, length, and lifetime. We explore the amount of bundling by measuring the number of microtubules/bundles in each frame (Figure 5C). We note that in TAMiT a detected microtubule means a line of fluorescence intensity, as shown by the colored lines in Figure 5B. This often represents a bundle with multiple microtubules.

In the *cut7-24* reference, there were, on average three microtubules/bundles per frame, with a broad distribution from 0 to 7 (Figure 5C). The mean lifetime was around 20 s, and the mean length was about 1.2 µm (Figure 5D–F). In *klp*6Δ cells, we find no significant differences in the average number of microtubule bundles compared to the *cut7-24* reference. As expected, the bundles were significantly longer, with a mean length of around 2 µm. Their lifetime was comparable to the reference (Figure 5C–F). This confirms that TAMiT can detect the previously measured phenotype of longer microtubules in *klp*6Δ cells.

In *alp*14Δ cells, based on previous work, we expect shorter microtubules due to defects in MT growth. Consistent with this, we found noticeably fewer detectable microtubules/bundles, with a mean below 2 and a maximum below 4 (Figure 5C). The smaller number of detected microtubule bundles is consistent with the idea that many microtubules are too short to be directly detected by TAMiT and instead are captured by the central Gaussian at the SPB. Microtubules in these cells are also shorter, with a mean length around 1 µm, and with a slightly decreased lifetime compared to the reference (Figure 5D–F). As for the *klp*6Δ cells, TAMiT analysis confirms the phenotype of *alp*14Δ.

In *cls*1-36 cells, Cls1p is inactive at the restrictive temperature. We find that this perturbation affects not only microtubule bundle number but also length and lifetime (Figure 5C–F). The mean microtubule bundle number is below 2: smaller than in the reference, and indeed is similar to *alp*14Δ cells. The upper range of the distribution is a bit larger than for *alp*14Δ, extending up to 5 bundles per frame. Of all the strains we analyzed, *cls*1-36 monopolar spindles showed the lowest lifetime with a mean of around 12 sec (Figure 5D). Despite this short lifetime, the mean length of microtubules/bundles was higher than in the reference and remarkably approached that measured in *klp*6Δ. We were surprised to find microtubules that were relatively long in *cls*1-36 cells since CLASP is thought to help stabilize bundled microtubules. A possible explanation comes from our observation that microtubules detected by TAMiT in *cls*1-36 cells were much lower in intensity, suggesting that these might be single microtubules. This suggests that in the absence of functional Cls1p, monopolar spindle microtubule bundling is greatly reduced, consistent with previous work [55]. The length distribution plots show the significant difference between microtubule/bundle length between *cls*1-36 and *alp*14Δ cells. About 50% of all microtubules in *cls*1-36 cells have length ≤1.75 µm compared to 1.0 µm for *alp*14Δ cells (Figure 5E). The distribution of microtubule lifetimes contains differences that are subtle (Figure 5F). For example, we observe that the curve increases steeply at a short lifetime for *cls*1-36 cells. This shows that in addition to a shorter mean lifetime, *cls*1-36 cells have a larger number of short-lived microtubules than in the other strains.

### 3.2. Dynamic Instability of Astral Microtubules in S. cerevisiae

In *S. cerevisiae* mitosis, long, curved astral microtubules can grow from the cytoplasmic face of the SPB (Figure 6A). Previous work suggested that astral microtubules are mainly single microtubules [43]. This makes them a good probe of microtubule dynamic instability and how it is affected by genetic background. However, quantifying astral microtubule dynamics is challenging because of their curvature, so they are typically laboriously hand-tracked [43]. We tested the ability of TAMiT to fit and track astral microtubules in 5 cells (Figure 6B). Fourier modeling of the microtubule path enables us to fit and measure the curvature (Figure 6C). We measured microtubule length by hand in ImageJ (Figure 6D, red) and also used TAMiT for automated measurement (Figure 6D, green). TAMiT captures the polymerization and depolymerization characterisitc of dynamic instability. As TAMiT fits the entire shape of the microtubule, it can also capture local and mean curvature (Figure 6E). Microtubule length measurements can then be used to extract polymerization and depolymerization speed (Figure 6F). When we compared the hand measurements against the automated results from TAMiT, we found no significant difference in measured speed. This suggests that TAMiT can automate the measurement of the dynamics of curved astral microtubules in *S. cerevisiae*.

## 4. Discussion

Advances in fluorescence microscopy have led to an explosion in the volume of image data [56,57]. Because manual analysis is and time-intensive, there is an increasing need for automated image analysis tools. Therefore, we developed TAMiT, to detect and track microtubules in *S. pombe* and *S. cerevisiae*. Related work has used similar techniques to ours, surface spline methods or spot tracking of tip-associated proteins [2,4,14,17,58] However, previous methods were limited in that they were semi-automated, lacking optimization, inapplicable to live-cell data, or restricted to two dimensions. Further, tracking microtubule tip-associating proteins typically cannot give full microtubule dynamics without inferences about non-imaged shrinking events. Building on previous work, TAMiT can overcome some of these restrictions. It is designed to work with fluorescently tagged microtubules, so can detect shrinking events. It runs unsupervised, optimizes parameters, tracks microtubules in 3D, and is designed for live-cell data from yeasts.

Biological structures are three-dimensional. Despite the ease with which fluorescence microscopy produces 3D image stacks, analysis difficulties mean that the Z dimension is commonly thrown out in favor of studying maximum-intensity projections. While this is justified in vitro where microtubules are often surface-bound, microtubule structures in cells are three-dimensional. As a result, ignoring the third dimension may introduce errors in the inferred microtubule dynamics. TAMiT overcomes this by treating microtubules as fully three-dimensional. TAMiT combines microtubule detection with optimization to yield fit parameters with uncertainties. This can improve the measurement accuracy compared to using only the estimated position.

A significant challenge in automated microtubule detection in cells is the typically low signal-to-noise ratio (SNR) used to avoid photodamage to the cell. By simulating images of microtubules at varying SNR, we find that TAMiT detects >95% of simulated microtubules at SNR ≥ 1.25, and achieves sub-pixel accuracy at SNR ≥ 1.5. This is possible because detecting a linear polymer allows information from multiple pixels to be used. When we tested TAMiT on experimental images, we quantified and confirmed previous results from perturbation to *S. pombe* microtubule dynamics, length, and number. In *S. cerevisiae*, TAMiT tracked single curved microtubules accurately and measured dynamics parameters similar to those found by hand tracking.

For future work, the structure of TAMiT is designed to facilitate other types of analysis. An inheritance-based format for the code allows one to easily add new Gaussian features to TAMiT without needing to overhaul the entire framework. Similarly, the modular representation of features as combinations of spots, lines, and curves allows the creation of different composite. However, TAMiT currently does not have an implemented graphical user interface, which means that the use of TAMiT requires basic MATLAB expertise to run the code. As we have shown, TAMiT can track microtubules in multiple phases of the cell cycle, and in both *S. pombe* and *S. cerevisiae*. Flexibility in the framework means that TAMiT is not specific to a single cell phenotype, and can be used to analyze other types of cells not considered here. However, the presence of multiple rounds of optimization and a large number of parameters per microtubule means that convergence to a solution in the non-linear optimization space can be slow. For example, for 3–5 features, analysis of a single frame in *S. pombe* takes 1–2 min, while a frame in *S. cerevisiae* takes 4–6 min. Therefore, tracking 100–1000 microtubules in a mammalian spindle is unrealistic for TAMiT in its current form. However, since TAMiT can handle curvature in microtubules, this leaves open the possibility of a future, accelerated implementation for mammalian cells.

Here, we have shown that TAMiT can accurately and automatically track microtubules in 3D in both *S. pombe* and *S. cerevisiae*. In future work, several extensions could further improve the utility of TAMiT. Calibrating the Gaussian intensity to the number of microtubules in a bundle as in previous work [59] would enable a better understanding of microtubule dynamics that vary due to bundling. Creating a GUI would improve access to TAMiT for researchers without a programming background. Finally, the larger goal is to eventually track all the microtubules in a mammalian spindle. To this end, speeding up the computation time for feature optimization would allow the tracking of more microtubules and pave the way for tracking a mammalian spindle.

## Figures and Tables

**Figure 1 biomolecules-13-00939-f001:**
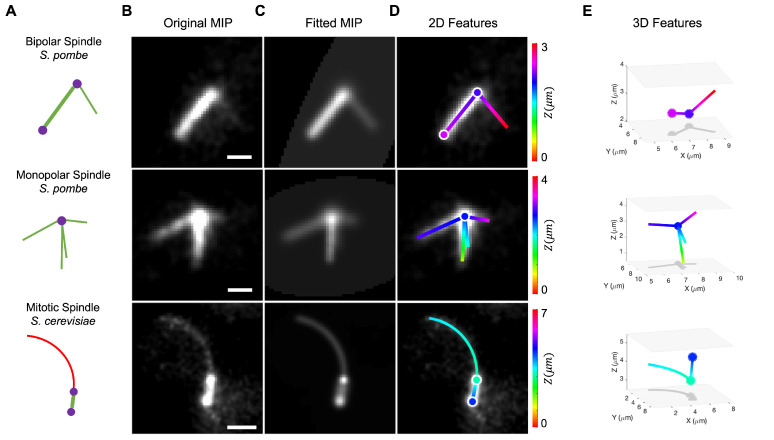
Feature detection and 3D fitting to various microtubule structures: a bipolar and a monopolar spindle from *S. pombe* and a spindle and astral microtubule from *S. cerevisiae*. (**A**) Schematics of three microtubule assemblies (bipolar spindle and monopolar spindle in fission yeast, and spindle with astral microtubule in budding yeast) and their representation as composite objects. Straight (green) and curved (red) microtubules are organized by spindle pole bodies (purple) (**B**) Maximum-intensity projection (MIP) images are created from image stacks of microtubule fluorescence. These images were filtered and contrast-enhanced to make dim microtubule bundles or single microtubules more visible. (**C**) Fitted intensity images created by TAMiT, displayed as a MIP. Each image is an output of TAMiT’s optimized 3D microtubule model. (**D**) Features determined by TAMiT, displayed as 2D projections from the optimized 3D model. Features (lines, curves, spots) are colored according to their position in Z and are overlayed on the original experimental images from (**B**) for comparison. (**E**) 3D visualization of features determined from the TAMiT model. Color represents the position in Z, as shown in the color bar on the left. (Scale bars: 1 µm).

**Figure 2 biomolecules-13-00939-f002:**
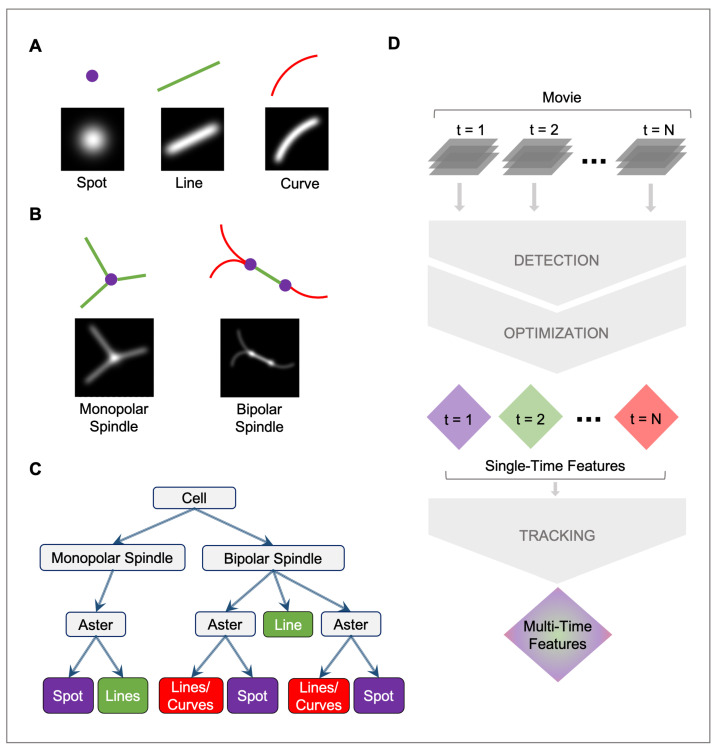
Schematic of image features, composite objects, and TAMiT workflow. (**A**) Schematic and MIP of the simulated intensity distribution, shown for the basic 3D elements, include a spot (representing an assembly of short microtubules, such as occur near a spindle pole), a line (representing a straight microtubule or bundle), and a curve (representing a curved microtubule or bundle). (**B**) Schematic and simulated MIP of composite images that can be constructed from spots, lines, and curves. A monopolar spindle contains a spot with one or more straight lines that end at the spot. A bipolar spindle contains two spots connected by a straight line. Lines or curves may extend from the spots. (**C**) A tree representation of the organization of microtubule structures and their connections in TAMiT. The monopolar or bipolar spindle resides inside the highest-level composite structure, a biological cell. (**D**) Flow diagram of TAMiT. A movie is analyzed frame by frame. Each frame undergoes a detection and an optimization step to yield features for that frame individually. Once all the frames are processed, the single-frame features are tracked to yield multi-time features. Tracking throws out any single-time features that may be spurious.

**Figure 3 biomolecules-13-00939-f003:**
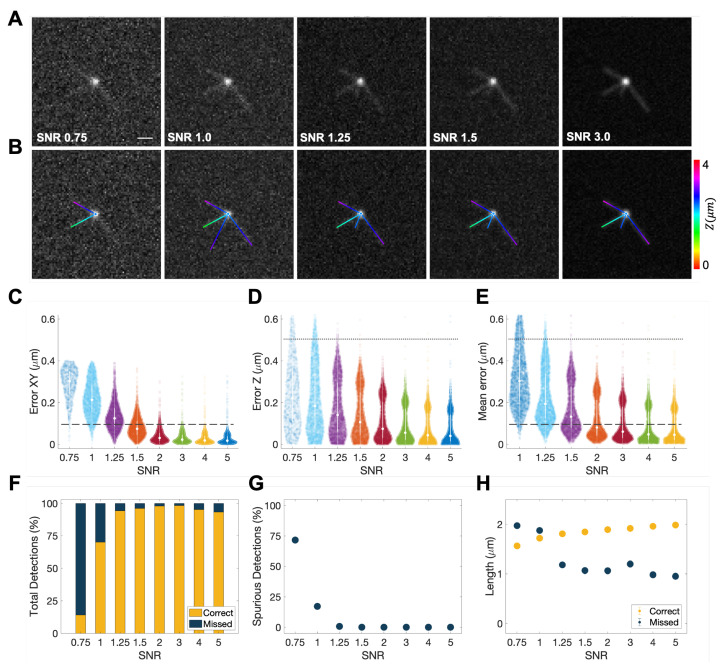
Accuracy of microtubule detection in TAMiT as a function of signal-to-noise ratio (SNR). We simulated 3D images of a monopolar mitotic spindle with varying SNR. (**A**) Simulated MIP image of the same monopolar spindle at SNR =0.75, 1.0, 1.5, 3.0, and 5.0. Scale bar, 1 µm). (**B**) Features detected by TAMiT from the images shown in (**A**), displayed as 2D projections. (**C**–**E**) Error in the spatial position of correctly detected microtubules as a function of SNR. As expected, the error decreases with increasing SNR. (**C**) In-plane (xy) position error. (**D**) Out-of-plane (z) position error. (**E**) Mean 3D error is sub-pixel for SNR >1.5. (**F**) Percentage of microtubules that were correctly detected and those that were missed by TAMiT. Correct detection percentage was low (15% and 70%) at low SNR (0.75 and 1.0). However, more than 90% of microtubules were correctly detected at SNR ≥1.25. (**G**) Percentage of spuriously detected microtubules versus SNR. These microtubules either did not exist, or their fitting error was too large. (**H**) Fitted length of microtubules for correct and missed detection versus SNR. The length of missed microtubules was smaller for higher SNR, suggesting that longer microtubules were less likely to be missed. The length of correct microtubules was higher at higher SNRs, suggesting that longer microtubules had a larger probability of detection.

**Figure 4 biomolecules-13-00939-f004:**
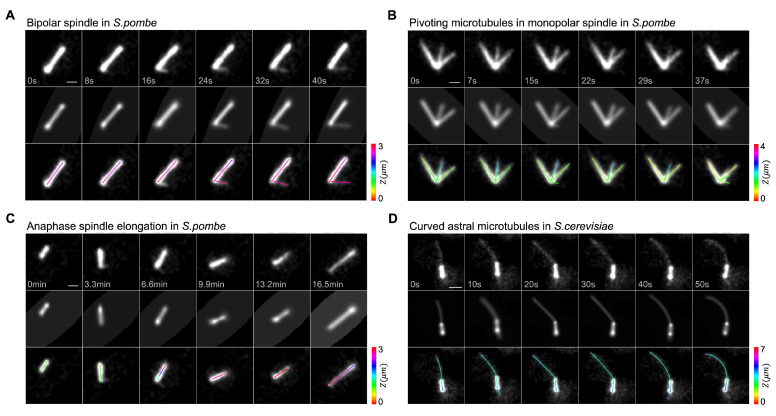
Robust detection of mitotic microtubule assemblies in yeasts. (**A**) Bipolar spindle in *S. pombe*. TAMiT detects both the spindle and a growing polar microtubule (last four frames) that grows away from the spindle at an angle. (**B**) Monopolar spindle in *S. pombe*. TAMiT detects rotating microtubule bundles (the central microtubule rotates clockwise between frames 1 and 5). (**C**) Elongating spindle in *S. pombe* anaphase B. TAMiT accurately detects the growing spindle. (**D**) Cruved astral microtubule in *S. cerevisiae* during anaphase spindle positioning. The long microtubule grows and bends upon interacting with the cell cortex. TAMiT detects these long dynamic microtubules with varying curvature. Scale bars, 1 μm. In each panel, the top row is a maximum-intensity projection (MIP) of the experimental microtubule fluorescence image stack. The second row is a MIP of the best-fit intensity image from TAMiT’s optimized model. The third row shows a 2D projection of the 3D features identified by TAMiT. The color along each feature shows the position in Z.

**Figure 5 biomolecules-13-00939-f005:**
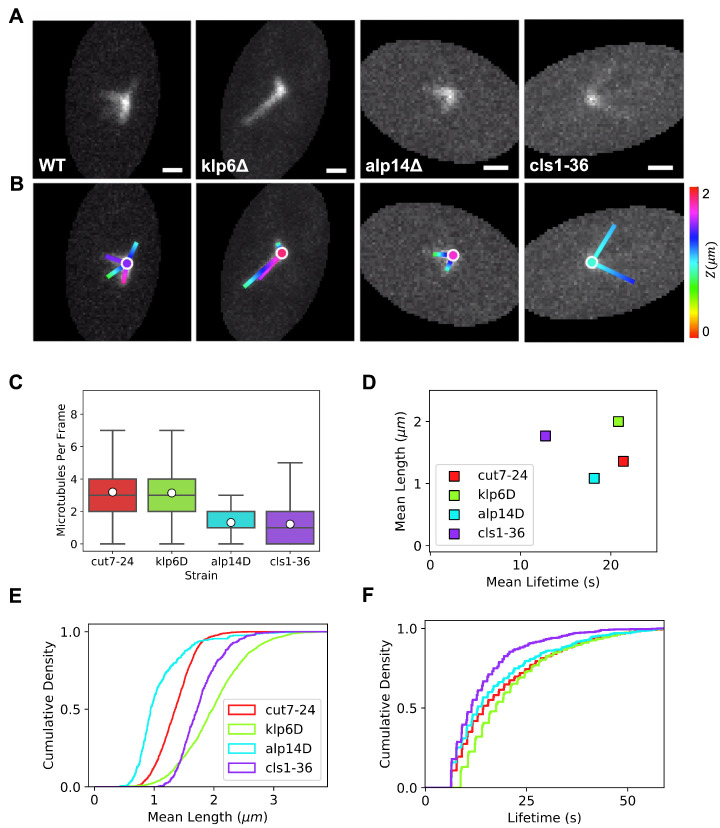
Quantification of monopolar spindle microtubules for *S. pombe* mutants that alter microtubule dynamics and bundling. (**A**) Maximum-intensity projection (MIP) of the experimental microtubule fluorescence image stack for *cut7-24* cells, with the additional perturbations *klp*6Δ, *alp*14Δ and *cls*1-36 as labeled. Scale bars, 1 µm. (**B**) Optimized features detected by TAMiT, displayed as a 2D projection overlaid on the experimental MIP image. (**C**) Box plot of the TAMiT detected number of microtubules/bundles in each image frame. (**D**) The plot of the mean microtubule lifetime as a function of the mean microtubule length, as measured by TAMiT in each genetic background. Error bars represent the standard error. (**E**) Cumulative probability density of microtubule/bundle length. (**F**) Cumulative probability density of microtubule/bundle lifetime.

**Figure 6 biomolecules-13-00939-f006:**
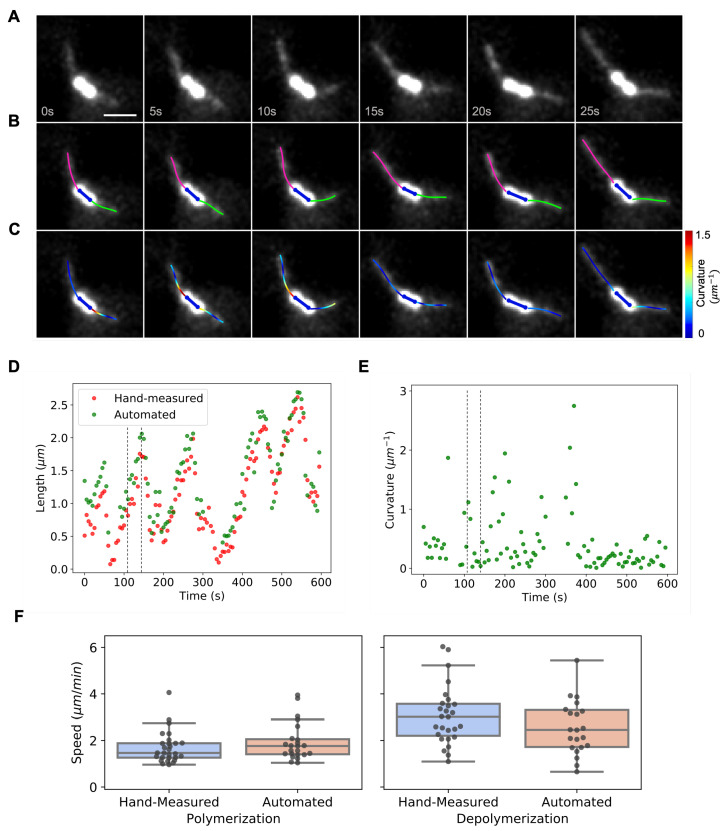
Measurement of astral microtubule length, dynamics, and curvature of in *S. cerevisiae*. (**A**) Maximum-intensity projection (MIP) of the experimental microtubule fluorescence image stack of a *S. cerevisiae* mitotic spindle with long, curved astral microtubules. Scale bar, 1 µm. (**B**) Optimized features detected by TAMiT, displayed as a 2D projection overlaid on the experimental MIP image. The spindle is colored blue, and the astral microtubules are pink and green. (**C**) Tracked microtubules are colored according to their local curvature (as shown in the color bar). Red indicates higher curvature and blue lower curvature. (**D**) Microtubule length as a function of time measured for an astral microtubule. The automated fit by TAMiT (green) is similar to the hand-measured values (red). The dashed lines indicate the time points shown in (**A**–**C**). (**E**) Mean curvature of the microtubule with length measurements shown in (**D**). (**F**) Dynamic instability parameters quantified for 5 astral microtubules. Data points represent polymerization and depolymerization events. Using hand measurement, we observed 28 polymerization and 27 depolymerization events. In comparison, TAMiT recorded 23 polymerization and 22 depolymerization events. Using a *t*-test, we failed to reject the null hypothesis of equal means (p=0.27 for polymerization and p=0.43 for depolymerization events).

## Data Availability

The code for TAMiT is available here: https://github.com/Betterton-Lab/TAMiT.git (accessed on 20 July 2022).

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
