# Peer review of "Quantifying Yeast Microtubules and Spindles Using the Toolkit for Automated Microtubule Tracking (TAMiT)"

_biomolecules, 2023, doi:10.3390/biom13060939_

Round 1
Reviewer 1 Report
In the manuscript entitled “Quantifying yeast microtubules and spindles using the Toolkit for Automated Microtubule Tracking (TAMiT)”, Ansari and coworkers present a software which detects, optimizes and tracks fluorescent microtubules with sub-pixel accuracy. Several applications were performed and validated the software for microtubules of yeast cells. Overall, I believe the manuscript is very well written and corresponds to a robust approach for improving microtubule fluorescence imaging quantifications. I suggest publishing the manuscript in its current format.
Reviewer 2 Report
In this manuscript by Ansari et al., the authors describe a novel program they developed to detect and quantify fluorescently labeled microtubules and mitotic spindles in 3-D time-lapse microscopy images. Overall, the work is interesting and clearly described, and will be of interest to the scientific community. Some revisions will be required to clarify some issues, provide some additional information, and clarify the advantages offered by this program over other image analysis tools.
Below is a list of issues that should be addressed prior to publication:
1. The authors mention multiple times that tools such as the one developed here are necessary because they eliminate the human bias and save time. Whereas I agree with this assessment, the information provided in the manuscript does not tell us whether this tool satisfactorily addresses the issues of bias and time. For instance, the data in figure 6D and F show that there is no difference between the hand and machine measurements. Therefore, there is no support here for the existence of human bias. However, the authors could make the point that these hand measurements were made by an experienced researcher and that using a software would allow researchers with minimal image analysis experience to make accurate measurements, without the steep learning curve the hand quantification typically requires. As for the time saving argument, nothing of what the authors say is very convincing. Indeed, the time taken by the program to analyze one video seems to be still quite long. However, the authors could provide information about how long it takes to make the same measurements by hand, so that the readers would have a reference. Moreover, they could clearly state that one major time-saving advantage of using a software is that one can queue many videos for analysis and then let the software run the analysis unsupervised.
2. The authors may want to explain early on (e.g., at the end of the introduction) that the tool they developed has machine learning capabilities. This aspect of the work may be intriguing for the readership because it means that the algorithm can “learn” to analyze structures/cells different from those analyzed here. In the current version of the manuscript, this only becomes clear later.
3. The authors mention that their tool is better than others that have been developed in the past, but this argument is always vague and somewhat superficial, with references all lumped together. It would be very helpful to the readership if they added a side-by-side comparison of this new tool with others. This could be done in the main text of the discussion and with the addition of a table.
4. Lines 26-29. It is unclear why the authors started their introduction with some very basic and elementary descriptions. In fact, the second sentence of the introduction is also inaccurate because tagged proteins will glow when illuminated with light of a certain wavelength, but this can be achieved in ways other than a laser. I believe that anyone who decides to read this paper will be already familiar with fluorescence microscopy. Therefore, I strongly advise the authors to eliminate some of the basic descriptions provided at the beginning of the introduction and perhaps expand on some concepts that are introduced later (e.g., the microtubule plus-end dynamics discussed in lines 62-63).
5. Lines 44-45. The authors may want to consider revising to avoid the use of the word “during” twice in a span of four words.
6. Line 48. Please, replace “there” with “their”
7. Lines 50-51. This is an incomplete sentence.
8. Lines 52-53. Please add some references to studies in which these tracking methods have been used.
9. Lines 55-57. It is unclear what point the authors are trying to make here. Also, even though the sentence starts with “However,” it seems totally disconnected from the previous sentence.
10. Line 90. Replace “though” with “through”
11. A lot of the information in the opening paragraph of Materials and Methods section does not really belong in the Materials and Methods and should be moved to the introduction.
12. Lines 95-96. The references to the figures are incorrect. It should be “…spindles from S. pombe (Fig. 1, top two rows; Movies 1-2), and from the mitotic spindle of S. cerevisiae (Fig. 1, bottom row; Movie 3).”
13. Figure 1 legend, second line. The word “and” is repeated twice.
14. Line 116. There is something wrong with this sentence.
15. Figure 2 legend, fifth line. Confusing sentence.
16. Line 121. “in this form” repeated twice.
17. Equation (3). Shouldn’t the subscript of parameter q be i instead of n? If so, make sure you also revise the text in the line below.
18. Line 124. The authors state that “using a Fourier series approximation with N = 2 was adequate.” It would be important for the authors to provide additional information to justify this statement. For instance, they could show the approximation results/fitting errors when N = 1, 2, 3, 4, etc. This information may be useful to readers who may want to use this software to analyze different cell types or microtubule structures.
19. Sections 3-5 of supplementary material. The authors mention a threshold, but do not provide information on how to identity/choose these threshold values, nor they provide information on the values used in this paper. Similarly to the case of the N value discussed in the comment above, these threshold values are hyperparameters needed to fine tune the program and may be useful to users interested in applying this tool to different types of samples. In general, adding more details on the “learning” steps of the model and justification for the chosen optimization algorithm could be helpful to potential users.
20. Line 136. It may be better to say “along the Z axis of the image.”
21. For all the numbers followed by a length unit, please make sure that there is a space between the number and the unit, but not between the two letters of the unit (this was a mistake that recurred many times throughout the manuscript).
22. Would it be possible to add error bark to some of the graphs that are currently missing them (e.g., Fig. 5D)?
23. Line 202. The wording “using live preparation” is a bit strange.
24. Line 247. There is something wrong with this sentence.
25. Figure 4 legend. There is something wrong with the sentence spanning the last and second-to-last line of the legend. And in the last line, it should be “each feature” instead of “each features”
26. Lines 329-330. There is something wrong with this sentence.
27. Line 332. The word “characteristic” is misspelled.
28. Figure 5. The top, left panel is labeled “WT.” But shouldn’t this be cut7-24? Also, could some data from manual tracking be added here as it was done for the budding yeast data in Figure 6?
29. Figure 6F. When discussing these results in the main text, the authors claim that there was no difference between the hand measurements and the automated measurements. Please, add p values to these graphs and information about the statistical test used. It is also unclear what the dots represent. This should be explained in the figure legend. Currently, the figure legend states that the data are from analysis of five microtubules. However, there are more than five dots, so they must correspond to something else. Please, add this information in the figure legend.
30. Figure 6 legend. Delete the word “of” the second time it appears in the first line of the legend.
